# Partner Selection Strategies in Global Business Ecosystems: Country Images of the Keystone Company and Partner Companies on the Brand Quality Perception

**Dongock Bang [1], Jiwon Lee [2] and Matthew Minsuk Shin [1,*]**

1   Department of International Trade, College of Social Science, Konkuk University, Seoul 05029, Korea; dongockbang@gmail.com
2   ELM Business School, HELP University, Bukit Damansara, Kuala Lumpur 50490, Malaysia; jiwon.lee@help.edu.my
*   Correspondence: shinm@konkuk.ac.kr; Tel.: +82-2-450-3774

**Abstract:** Consumers perceive brand quality from the country of origin of the brand. Global business ecosystems represent multiple countries such as the country of the keystone company and the country of the assembly companies. Thus, the brands of global business ecosystems have multiple countries of origin. This study aims to examine the impacts of the country images of the keystone company and assembly companies on consumers' brand quality perceptions. In addition, depending on the assembly partner selection strategies of forming a global business ecosystem, the characteristics of the associated countries with the business ecosystem may change. The keystone company may select an assembly partner from a developing country or from a developed country. These two cases are compared to examine the impacts of the combined country images of the keystone and assembly companies. To do so, this study surveys Vietnamese consumers' perceptions of the brand Hyundai Motor, the country images of South Korea as the country of the keystone company, India as the assembly partner from the developing country, and USA as the assembly partner from the developed country. The collected data were analyzed using a structural equations modeling method and results are discussed with theoretical and managerial implications.

**Keywords:** global business ecosystem; keystone country company; cognitive country image; affirmative country image; sustainability; downward and upward assembly



## 1. Introduction

The business competition has evolved from the competition between one company and the other companies to the competition between a business ecosystem and other business ecosystems [1]. Over the past two decades, many studies focused on the business ecosystem level competition. For example, applied strategy research has increasingly focused on co-dependent system of complimentary firms, through concepts such as ecosystem [2], industry architecture [3], and platforms [4].

One of the competitiveness measures of the business ecosystems is consumers' preference of the brand produced by the ecosystem [5,6]. For example, consumers may have positive or negative quality perceptions from the brand Hyundai Motors, which is a brand produced by a global business ecosystem with Hyundai being the keystone company [7]. The quality perception depends on price, design, heritage, and many other attributes about a brand [8]. One of these attributes is the country of origin (COO) and the country image that the origin represents [9]. In the COO perspective, consumers may perceive multiple country images from a brand produced by a global business ecosystem as the ecosystem is comprised of multiple countries including the country of the keystone company, the country of the assembly company, and other countries involved in the business ecosystem [10]. Thus, consumer perception on the quality of a brand produced by a global

business ecosystem may be influenced by multiple country images associated with the ecosystem [11].

In this context, Apple, Samsung, Tesla, and other global brands created by the global business ecosystems have multiple COOs. For example, the famous inscription on the back panel of an iPhones reads "*Designed by Apple in California. Assembled in China*". This tells us that an iPhone has two COOs, USA (California) as the country of the keystone company (COK) and China as the country of assembly (COA). COK, also referred to as brand origin, is the country that consumers emotionally associate with a brand as its origin [12]. COA is the country that actually performs the final assembly of the branded product [13]. Thus, consumers may perceive quality of Apple from country images of both COK and COA.

Therefore, in COO perspective, consumers' quality perception toward a brand may depend on different strategies in selecting the global business ecosystem partners: in particular, the assembly partner. The first type of assembly partner selection strategy is downward assembly, which illustrates when the COA has lower country image than that of the COK [14]. The second type, upward assembly, occurs when COA has higher country image than that of the COK [14].

In this context, the current study examines consumers' quality perception of the brand Hyundai Motor and compares the influences of the country images of South Korea (COK), the USA (COA of upward assembly), and India (COA of downward assembly) on consumer's brand quality perception.

The remainder of this study is organized as follows. Section 2 reviews the relevant literature. Section 3 describes the research method and Section 4 presents the analysis results. Section 5 offers theoretical and managerial implications based on our findings.

## 2. Literature Review

### 2.1. Partner Selection

Partner selection is one of the important factors in forming a competitive global business ecosystem. A group of studies on the partner selection, e.g., [15–18], divided the selection strategy into two categories based on partners' (1) requirements and (2) collaborative performance. First, the requirements of the partner are further defined into teamwork skills, knowledge sharing, communication, and problem-solving ability. Second, cooperative performance is referred to as cooperation between companies in the past, which leads to mutual understanding and cohesiveness, while reducing uncertainty and conflict. In addition, another group of scholars, e.g., [19,20], noted that different partner selection mechanisms play an important role in the competitiveness of the ecosystem. These studies also report the importance of resources and capabilities shared through appropriate cooperation with the partners.

Reviewing exiting studies, it can be summarized that selecting the right partners among those with diverse resources and capabilities may increase competitiveness of the entire business ecosystem.

### 2.2. Global Business Ecosystem

The business ecosystem is an interdependent community that expands the traditional supply chain partners by involving more stakeholders such as universities, government, and industry associations in the network [21]. All of the stakeholders share a common vision and fate by contributing their complementary resources and capabilities in order to create a new business project or an emerging industry [2].

In the business ecosystem perspective, brand companies act as the keystone companies while other parts and service providers are considered the partner companies [22]. In this context, the global business ecosystem is comprised of a keystone company and many international partner companies [3]. The automobile industry is often discussed as an example of such a global business ecosystem with the keystone company (a brand such as Volkswagen) orchestrating multiple different partner firms all over the world to assemble, market, and distribute the end production [23].

### 2.3. Country-of-Origin Images

One striking phenomenon in the past two decades has been the increase in the brands produced by global business ecosystems that are assembled in diverse locations around the world and marked under a single brand name [24]. Such brands are also referred to as multinational products [24].

Consumers often evaluate products based on the countries from which they originate [25]. One of the major informational cues for consumer perceptions is the images of the country of origins (COOs) represented by multinational products [24]. For multinational products, the country image is composed of various attributes such as the country of keystone company (COK) and country of assembly (COA) [26]. In short, in the case of multinational products, the COO can be decomposed into the "designed by" (COK) and the "assembled by" (COA) countries.

Consumer perceptions may respond to COK and COA cues separately and differently, rather than simply responding to an overall COO [27]. Thus, COK and COA images of a given multinational product might influence consumers' perceived product quality differently [27].

The image of the COK is associated with a product as an institutional support such as planning, design, and marketing [28]. For multinational products, the country image of COK amounts to brand perceptions, which help consumers evaluate the product quality and make purchasing decisions [29]. For instance, New Zealand milk brands are associated with high quality with the institutional image of New Zealand, which includes strict quality control, a transparent system, good business practices, and ever-improving environmental management [30].

The country image of the COA is also used by consumers to judge the quality of the product as COA information is most visible to consumers through the "Made in" labels [26]. The majority of consumers prefer products assembled in developed countries as they are considered of high quality while products from less developed or developing countries are perceived of lower quality [31].

### 2.4. Cognitive and Affective Country Images of COK

Country image (CI) is not merely a cognitive cue for product quality, but also relates to emotions, identity, pride, and autobiographical memories [32]. In this view, CI can be examined in terms of two dimensions: (1) a cognitive dimension and (2) an affective dimension. Cognitive country image (CCI) includes technological advancement and the country's industrial development and technological advancement, while affective country image (ACI) describes consumers' affective response to the country's people, culture, history, etc. [33].

Halo effect is one of the ways to explain the impacts of CCI and ACI on consumers' perceived quality of a given product. Halo effect is originally discussed in the field of applied psychology, describing indirect effect on beliefs [34]. In the field of consumer behavior, halo effect can explain the impact of country images when a consumer is unfamiliar with certain foreign goods, and the CCI and ACI are often used to assess the brand quality of the product [35]. Thus, consumers are influenced by CCI and ACI of the CI of a given product on their perceived brand quality (PBQ) through halo effects of the associated country images. Based on above discussions, this study suggests following hypotheses:

**Hypothesis 1 (H1):** *ACI of COK has a positive impact on PBQ.*

**Hypothesis 2 (H2):** *CCI of COK has a positive impact on PBQ.*

### 2.5. Downward and Upward Assembly

Depending on the level of development (i.e., economic, political, cultural, etc.) of the COA, consumers may have different PBQ [36]. For example, consumers may perceive brand quality of Volvo differently when it is assembled in Sweden compared to in China [37].

Developed countries may offer higher brand quality while developing countries as the COAs may be associated with lower PBQ. Lee and Shin [14] define such assembly partner selection strategies, respectively, as downward and upward assembly. Downward assembly (DWA) refers to the business ecosystem selecting an assembly partner from a less developed country than the COK for the benefit of lower assembly costs [14]. Upward assembly (UWA) refers to the business ecosystem selecting an assembly partner from a more developed country than the COK to dilute the low development images of the COK [14].

Thus, when consumers evaluate the level of development of COK, which is associated with a given brand, they may activate stereotypical beliefs about the more developed country and retrieve brand quality from the country image. In other words, in the case of DWA, consumers mays perceive lower PBQ than in the case of UWA by the combined country images of COK and the assembly partner country from a developing country. On the other hand, in the case of UWA, consumers mays perceive higher PBQ than in the case of DWA by the combined country images of COK and the assembly partner country from a developed country. Therefore, this study proposes the following hypotheses:

**Hypothesis 3a (H3a):** *The combined ACI of COK and COA in the case of UWA has a stronger impact on PBQ than in the case of DWA.*

**Hypothesis 3b (H3b):** *The combined CCI of COK and COA in the case of UWA has a stronger impact on PBQ than in the case of DWA.*

*2.6. Research Model*

The relationships among the factors discussed above are organized into hypotheses and presented in a research model in Figure 1.

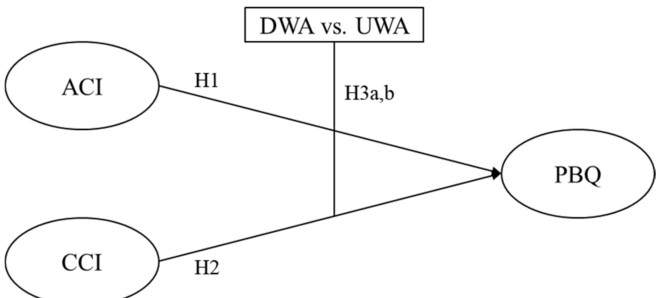

**Figure 1.** Research model and proposed hypotheses. Note: COK (country of keystone company), ACI (affirmative of country image), CCI (cognitive of country image), DWA (downward assembly), UWA (upward assembly), PBQ (perceived brand quality).

## 3. Method

*3.1. Measurement Items*

The survey is designed to assess consumers' ACI and CCI of COK (South Korea) and the two COAs (India and the USA). The last consists of questions to measure the PBQ by consumers. India and the USA were selected as the COAs to examine how Vietnamese consumers evaluate Hyundai Motor's brand quality according to the type of global business ecosystem partner selection strategies (downward and upward).

The reasons for conducting a survey of Vietnamese consumers' perceptions on Hyundai Motors are as follows. First, the sales volume of Korean automobiles, especially Hyundai Motors, which is the subject of this study in Vietnam, ranks first and has the highest recognition, and most of the respondents owned Hyundai Motors or had experience working with Hyundai Motors (see Table 1). The higher the awareness of the product, the more accurate the evaluation [38]. Therefore, Vietnamese consumers will be able to evaluate the Hyundai Motor brand in a relatively accurate manner.

**Table 1.** Vietnam car sales in the first quarter of 2021 [39].

| Rank | 1 | 2 | 3 | 4 |
|---|---|---|---|---|
| Brand | Hyundai | Toyota | Mitsubishi | VinFast Fadil |
| Sales Volume | 8007 | 6839 | 4602 | 4148 |

In addition, the reasons for selecting India and USA as the COAs are as follows: in terms of Vietnam's per capita GDP, it is in between the USA and India. This makes it possible to study what the quality perception of Hyundai Motor is according to upward and downward assembly.

Each question was measured on a 5-point Likert scale, and the average of responses to the grouped questions for each variable was used. A higher score is interpreted as a positive answer. The measurement items and the sources are presented in Table 2.

**Table 2.** Measurement items and references.

| Constructs | Measurement Items | References |
|---|---|---|
| Affective country image (ACI) | ACI1: (Country) is peace-loving.<br>ACI2: (Country) is friendly.<br>ACI3: (Country) is cooperative.<br>ACI4: (Country) is likable. | Li et al. [40] |
| Cognitive country image (CCI) | CCI1: (Country) is affluent.<br>CCI2: (Country) is economically developed.<br>CCI3: (Country) has high living standards.<br>CCI4: (Country) is advanced science and technology.<br>CCI5: (Country) is good living conditions. | |
| Perceived brand quality (PBQ) | PBQ1: The likelihood that the brand would be reliable is high.<br>PBQ2: The workmanship of the brand would be high.<br>PBQ3: This brand should be of good quality.<br>PBQ4: The likelihood that this brand is dependable is high.<br>PBQ5: This brand would seem to be durable. | Dodds et al. [41] |

### 3.2. Data Collection

In this study, online and face-to-face surveys were conducted for about two months from 1 July to 30 August 2021 for Vietnamese customers who live in Ho Chi Min. The questionnaire was divided into two types: English and Vietnamese. For translation of the questionnaire, English to Vietnamese were translated from Vietnamese who can speak English and Vietnamese. On the other hand, the correct translation was confirmed into English, and it was checked whether the translation from English to Vietnamese resulted in the same result. About 247 questionnaire responses were collected.

### 3.3. Analysis Method

In this study, the collected data were analyzed using the structural equation modeling (SEM) method using SPSS 25.0 statistical package and AMOS 24.0. The socio-demographic characteristics of respondents were analyzed by frequency analysis technology and statistical analysis. In the current paper, the validity and reliability of the measurement items were confirmed using reliability and factor analysis. To test the hypothesis, this study applied SEM to determine the relationships among variables as SEM is more robust in determining relationships between latent variables compared to linear regression [42]. Prior to verification, this study checked for normal distribution and the fit of the model to see if it was an acceptable model. The maximum likelihood estimation and the covariance approaches were used for the SEM analysis.

## 4. Empirical Analysis

### 4.1. Sample Characteristics

The general characteristics of the sample are: As for the gender, 73.3% of the respondents were male (181 people) and 26.7% (66 people) were female. As for the age, 108 respondents were in their 30s, which showed the highest distribution among the entire sample (43.7%), followed by 74 in their 40s (30%), 37 in their 20s (15%), and 28 respondents in their 50s (11.3%). In terms of monthly income, most of the 247 respondents received a monthly salary of USD 500–999 (217, 87.9%).

### 4.2. Correlations among Variables

The results of correlation analysis among variables are shown in the following Table 3. First, the correlations between PBQ and exogenous variables were all analyzed to be significant. Specifically, the relationship between CCI and PBQ was .510 ($p < .01$), showing the highest correlation. The correlation between ACI and PBQ was also high at .460 ($p < .01$).

**Table 3.** The results of correlation analysis among variables.

|  | **ACI** | **CCI** | **PBQ** |
|---|---|---|---|
| ACI | **.654** | | |
| CCI | .460 ** | **.562** | |
| PBQ | .279 | .510 ** | **.894** |
| Mean | 3.3613 | 3.6146 | 3.3020 |
| Standard deviation | .727 | .740 | .643 |

Note: Bold number shows the square roots of AVE for that construct, ** $p < 0.01$, ACI (affirmative of country image), CCI (cognitive of country image), PBQ (perceived brand quality).

### 4.3. Reliability and Validity

In order to verify the reliability and validity of the measurement items, this study conducted a confirmatory factor analysis (CFA) (see Table 4). The CFA shows that the explanatory power of the measurement items to explain the corresponding conceptual variable is good. Specifically, it can be seen that the t value of the coefficient of each measurement item is 13.268 or higher, indicating that the concentration validity is secured. In addition, it was also confirmed that the composite reliability (CR) and the average variance extracted (AVE) were both 0.7 or more and 0.5 or more, respectively, and are within the standards for reliability. Meanwhile, this study used correlation analysis to verify discriminant validity. As shown in Table 3, as a result of comparing the square root value of the AVE and the adjacent correlation coefficients, it was confirmed that all the square root values of AVE were high, indicating that there is no problem in discriminant validity.

In addition, this study used Harman's single factor verification method to solve common method bias that may appear in the survey technique. As a result of the analysis, it was identified that the factors with an initial eigenvalue of one or more were classified into three, and it was confirmed that there was no problem in the same method convenience.

### 4.4. Model Fit

The model fit analysis results are shown in Table 5 below. The Diamantopoulos et al. [43] study was referred to as the basis for judging the suitability of the model, and as a result of analysis based on this, the GFI and AGFI values were slightly below the standards. However, since all other indicators other than these two fall under acceptable categories, the SEM presented in this study is judged to be an acceptable model.

**Table 4.** Confirmatory factor analysis (CFA).

| Construct | Items | Std. Loading | Std. Error | t | Cronbach's a | CR | AVE |
|---|---|---|---|---|---|---|---|
| ACI | ACI1 | .855 | - | - | .976 | .944 | .809 |
| | ACI2 | .852 | .055 | 17.413 | | | |
| | ACI3 | .932 | .057 | 19.607 | | | |
| | ACI4 | .725 | .034 | 13.268 | | | |
| CCI | CCI1 | .849 | - | | .913 | .937 | .750 |
| | CCI2 | .754 | .052 | 13.959 | | | |
| | CCI3 | .919 | .051 | 19.162 | | | |
| | CCI4 | .796 | .057 | 15.157 | | | |
| | CCI5 | .805 | .05 | 15.444 | | | |
| PBQ | PBQ1 | .948 | - | - | .894 | .979 | .946 |
| | PBQ2 | .984 | .022 | 40.587 | | | |
| | PBQ3 | .962 | .025 | 35.323 | | | |
| | PBQ4 | .919 | .032 | 28.534 | | | |
| | PBQ5 | .903 | .032 | 26.747 | | | |

Note: ACI (affirmative of country image), CCI (cognitive of country image), PBQ (perceived brand quality).

**Table 5.** The model fit analysis results.

| Index | $\chi^2$/df | GFI | AGFI | CFI | NFI | IFI | RMSEA |
|---|---|---|---|---|---|---|---|
| Standard | 3$\geq$ | 0.9$\leq$ | 0.8$\leq$ | 0.9$\leq$ | 0.9$\leq$ | 0.9$\leq$ | 0.08$\geq$ |
| Results | 2.444 | 0.835 | 0.792 | 0.933 | 0.915 | 0.933 | 0.074 |

*4.5. H1 and H2 Results*

H1 and H2 results are described in Figure 2. It was found that CCI of COK had a significant effect on PBQ as 0.452 ($p < .01$). However, ACI of COK did not appear to have a significant effect on PBQ. Through this, it was found that only H2 was accepted while H1 was rejected.

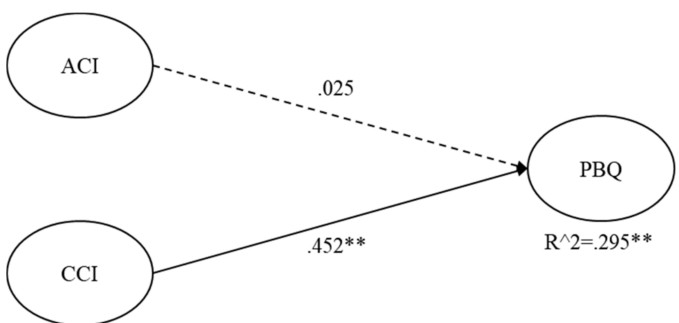

**Figure 2.** H1 and H2 results. Note: $p > 0.01$ **, ACI (affirmative of country image), CCI (cognitive of country image), PBQ (perceived brand quality).

*4.6. Multigroup Comparison: H3a/b*

Multigroup analysis was performed to determine the difference in coefficients between routes according to the country of assembly. It was confirmed that there was no significant difference in the samples when comparative analysis was performed on 133 in India and 114 in the US out of a total sample size of 247e According to the results of examining the fit of two independent structural equation models before performing the comparison, the GFI value and NFI were slightly insufficient compared to the reference value, but other indicators were good and thus it was judged to be an acceptable model.

After that, the $\chi^2$ value was derived by estimating the cross-group equality constraint model of the structural equation model for each group; $\Delta\chi^2$ between Model 1 and Model 2 is 5.2, which is less than 5.99, indicating that it is not statistically significant. This shows that there is no problem with the factor loading for measurement tools such as questionnaires. In the case of Model 3, there is a significant difference when compared with the non-constrained model (Model 1), but since this part is converted into a causal relationship corresponding to the hypothesis when it is converted to the structural equation model, the difference is not a big problem.

Next, the significance of the difference was verified by comparing the values of the coefficients appearing in each path below (Table 6). The ACI → PBQ pathway was not statistically significant (C.R. = 0.777), but it was statistically significant between the CCI → PBQ pathways (C.R. = 2.185). In particular, CCI → PBQ was higher when the country of assembly was the United States, and it was found that the path coefficient value fell when the country of assembly was India.

**Table 6.** Comparing the values of the coefficients.

| Path | Standard Coefficient | | C.R. |
| --- | --- | --- | --- |
| | DWA India and South Korea | UWA USA and South Korea | |
| ACI → PBQ | .029 | .187 * | 0.777 |
| CCI → PBQ | .117 * | .457 ** | 2.185 |

Note: ** $p < .01$, * $p < .05$, C.R. > ±1.965, ACI (affirmative of country image), CCI (cognitive of country image), PBQ (perceived brand quality).

## 5. Discussion

### 5.1. Theatrical Implications

First, this study showed that the CCI of COK had a positive impact on consumers' product quality perception. This result confirms the findings in existing studies such as Karami et al. [32] and Li et al. [40], which show positive associations between CCI and consumers' quality perceptions. Therefore, our empirical investigation adds an evidential support for existing theories. Second, as part of the global business ecosystem strategy, the current study suggested the different impact on consumer perceptions depending on which country the ecosystem partners with. Such expands existing discussions on multinational corporation (MNC) research such as Parida et al. [23], which looks at how MNCs should respond to heterogeneous markets. A heterogeneous market is a concept that encompasses partner companies and consumers. In the above two studies, the company itself constituting the ecosystem was regarded as an important factor. However, it overlooks the point that consumers evaluate the perceived quality of products differently depending on the country in which the company is located. In other words, consumers evaluate products differently depending on how the keystone company's country and the partner company's country build a combination. Exiting studies considered the company itself that constitutes the ecosystem as an important factor in common. However, it is overlooked that the national image of a company is an important factor in recognizing the quality of products. In other words, consumers perceive products differently depending on whether the keystone company country and partner company country are UWA or DWA. Such analysis broadens the horizon of the national image theory and fills the gap in the business ecosystem theory.

### 5.2. Managerial Implications

Based on the results of the above study, the following implications can be drawn. First, considering that COK is economically developed, the quality of life is high, and it has advanced science and technology, the brand quality is viewed as positive. On the other hand, it was found that COK's peace-loving, friendly, cooperative, and likable emotional

image had no effect on brand quality. It can be seen that the halo effect theory, which assumes the national image as a component and affects products, is partially supported. In other words, when the national image is made up of various components, it is possible to examine the influence of the national image on the quality of products in more depth. In terms of practical implications, it can be said that marketing that emphasizes CCI in product marketing raises awareness of product quality. As in the case of this study, when Korean automobiles are sold to Vietnamese consumers, it is expected that positive marketing effects will be obtained by emphasizing that the product is produced in a country that has developed economically and has advanced technology. Recent IPO of Volvo in Stockholm listing instead of Shanghai listing is a case that the finding of the current study can be used to analyze [44]. Volvo a Swedish brand which was purchased by China in 1999, still manufactures in Sweden and seeks to maintain its IPO in Sweden instead of China. According to the current study, these can be interpreted as Volvo's efforts to maintain higher brand quality perception by maintaining the COK and COA images within more advanced country of Sweden than China.

Second, upward assembly, which produces products in COA, which is more advanced than COK, has a positive effect on recognizing brand quality. This is a strategy for selecting partner countries such as production in the global business ecosystem theory, and it can be seen that the quality of products is evaluated more positively when produced in developed countries. This takes a different view of the typical overseeing product approach, which focuses on price competitiveness by using cheap labor to produce goods. In other words, if the perception of brand quality is more important than price competitiveness according to the characteristics of the product, a partner selection strategy called upward assembly should be selected. In practice, when price is an important factor in product selection, downward assembly might benefit the business. On the other hand, when brand quality is more important than price in product selection, production partners might be a better strategy based on upward assembly method. This implication can be evidenced in existing practices of the current automobile industry. Existing auto manufacturers are continuously launching electric vehicles (EV) and their partnerships with EV battery manufacturers differ depending on the price sensitive or luxury auto segment of the brand. Most price sensitive auto brands such as Toyota, Ford, Tata, etc., form battery manufacturer partners with Chinese companies while Porsche, which is a luxury sports car brand, maintains its battery partner in Germany [45].

*5.3. Limitation and Future Research*

The limitations of this study are as follows. First of all, the elements constituting the national image were considered only as cognitive and affective country image. There are various elements that make up the national image, and it is judged that the quality evaluation of automobile products will also appear differently. Accordingly, it is necessary to examine the influence of various components on the quality of automobiles in evaluating the brand of a product in the future.

Second, in understanding the global business ecosystem, it is necessary to examine whether automobiles produce the same results in partner selection strategies by comparing products such as computers and mobile phones. In other words, price competitiveness may be more important than quality for a specific product, so it is also important to analyze the impact of each product. Finally, this study examined how the national image of a product affects the quality evaluation. However, the national image of the product category may be more important than the national image of the product. Therefore, product categories can be an important factor in country selection in a global partnership strategy.

Third, the data studied in this article come from 247 questionnaire surveys. Although the current study confirmed reliability and validity of the data collected, future research with the research model is suggested with larger sample sizes needed to ensure the authenticity of the data.

Fourth, as the variables discussed in this study are country images and brand value, there are many other variables that could influence respondents' perceptions. Therefore, collecting data on more control variables is suggested for future research to ensure the feasibility of the model.

### 6. Conclusions

The current study examined how CCI and ACI of COK affect PBQ. Table 7 below summarizes the results of the hypotheses according to the results the current study.

**Table 7.** The results of the hypotheses.

| Hypotheses | Result |
| --- | --- |
| H1 | Rejected |
| H2 | Supported |
| H3a | Rejected |
| H3b | Supported |

As a result, the effect of CCI on PBQ was confirmed. However, the effect of ACI was not confirmed. Second, it was examined whether there was a difference in the combined country images on PBQ by the different formation of assembly partners. In the relationship between COK and COA, it was divided into a global business ecosystem consisting of a country where COK is more advanced than COA (DWA) and a global business ecosystem consisting of a country where COA is more advanced than COK (UWA). As a result of this, in the case of UWA, it showed a stronger impact of the combined CCI of COK and COA on PBQ than that of DWA.

**Author Contributions:** Conceptualization, M.M.S.; methodology, J.L.; writing—original draft preparation, D.B. and M.M.S.; writing—review and editing, M.M.S.; project administration, M.M.S. All authors have read and agreed to the published version of the manuscript.

**Funding:** This research received no external funding.

**Institutional Review Board Statement:** The authors of the current study obtained the IRB approval for the methods used in this study from the Konkuk University IRB.

**Informed Consent Statement:** Not Applicable.

**Data Availability Statement:** No new data were created or analyzed in this study. Data sharing is not applicable to this article.

**Acknowledgments:** Authors of the current study, D.B., J.L. and M.M.S. are grateful for the anonymous peer reviewers and the editors for their critical comments, which helped to significantly improve the quality of this paper. All authors have read and agreed to the acknowledgement.

**Conflicts of Interest:** The authors declare no conflict of interest.

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
