# Peer review of "Partner Selection Strategies in Global Business Ecosystems: Country Images of the Keystone Company and Partner Companies on the Brand Quality Perception"

_sustainability, doi:10.3390/su132212903_

Round 1

Reviewer 1 Report

I would like to congratulate the authors for their interest in researching in this field, however, the work presented presents some deficiencies.

  1. a) Wording of the title should be revised. The use of ":" is not appropriate in this case and we consider that it should therefore be eliminated.

Thus, the title would read as follows: “Partner selection strategies in global business ecosystems based on the country images of the keystone company and  partner companies”.

I also recommend that the authors take into account the following aspects in the event that they decide to reform this title.

The title should have the following characteristics:

-Describe the content of the article in a specific, clear, accurate, brief, and concise manner.

-Enable the reader to identify the topic easily.

-Allow a precise indexing of the material.

  1. b) Abstract does not provide information for each part of the work, so it would not comply with the fundamental information of a research summary.

More specifically, you should briefly explain the methodology used to process the study data as well as a summary of the results obtained.

  1. c) The introduction should not include aspects referring to the research itself and its development. That is why the last paragraph of the introduction should be reformed.
  2. d) The description of the survey shown in section 3.1 is rather confusing and difficult to understand and is not sufficiently developed to fully understand the article.

I would recommend that this section be rewritten to include, if necessary, lists or enumerations to help understanding.

Authors should review this section and indicate in detail the survey used.

  1. e) 3.3 Analysis method - authors have not indicated the statistical methods used and their justification. These methods should be indicated in detail.
  2. f) There is no discussion section. This section, independent of the conclusions section, is very important to demonstrate the scientific interest of the proposed article and its degree of association with other previously published sources.

I recommend that the authors include this section before the conclusions chapter.

  1. g) Conclusions chapter is correct, although they should be based on the reflections in the Discussion section. For this reason, after the reform of this section, the Conclusions section should be revised.

In addition to the proposed changes, the article requires a thorough revision regarding its grammar and general writing (especially in the use of punctuation marks).

I hope that these changes will help to improve your article and make it a document of great scientific interest.

Reviewer 2 Report

Dear authors,

Thank you for your paper. It was enjoyable to read, however I have an observation I want to adress you:

The references are too old/outdated, only 17 of 44 references are from 2010 until now, so References section should be updated with recent sources.

Good luck,

Reviewer 3 Report

The research in this paper points out that in order to improve brand quality in the global business ecosystem, not only the national image of the country where the keystone company is located, but also the national image of the country where the partner company is located must be considered. The research perspective is novel, but there are still the following problems:

  1. The innovative points of the issues discussed in this article are not very novel, and the conclusions drawn are not of much guiding significance for practical issues.
  2. There are problems with the structure of this article. The second part of the literature review starts from the business ecosystem, the image of the country’s origin, the cognitive and emotional country image of COK and COA, the control method: downward and upward assembly, and the research model. However, many of the discussions do not belong to the literature. The content of the review.
  3. The problem studied in this paper is the influence of the national image of the key company on the brand quality, and the adjustment of the national image of the cooperative company on the brand quality when the cooperative company is selected upward or downward. The key argument is "brand quality", but the title of the article does not mention "brand quality".
  4. The related works about Partner selection should be further reviewed, such as: The partner selection modes for knowledge-based innovation networks: A multiagent simulation. IEEE Access, 7, 140969-140979; A method of partner selection for knowledge collaboration teams using weighted social network analysis. Journal of Intelligent Systems, 27(4), 577-591.
  5. The abstract is not smooth and clear enough. There is no concise and concise point out the problems to be studied in this article, nor does it point out the innovative points of this article. It is necessary to reorganize the language of the abstract part.
  6. The data studied in this article comes from 247 questionnaire surveys. A single data source is difficult to guarantee the reliability of the data. Because the authenticity and reliability of the data obtained from the questionnaire survey is low, the sample needs to be further expanded to ensure the authenticity of the data .
  7. The variables discussed in this article are national image and brand value, but for practical issues, there are many other variables involved. Therefore, more control variables need to be considered in the established model to ensure the feasibility of the model.

Based on the above comments, it is recommended to review carefully after careful revision.

Round 2

Reviewer 1 Report

Authors have done a good job and have responded clearly and precisely to the suggestions made. 
I trust that these modifications will serve to increase the scientific interest of the article.

Reviewer 3 Report

The authors have revised the paper carefully. So I suggest to accept it